# Toward scalable biocatalytic conversion of 5-hydroxymethylfurfural by galactose oxidase using coordinated reaction and enzyme engineering

William R. Birmingham [1], Asbjørn Toftgaard Pedersen [2,4], Mafalda Dias Gomes[2], Mathias Bøje Madsen[2], Michael Breuer[3], John M. Woodley [2] & Nicholas J. Turner [1✉]

5-Hydroxymethylfurfural (HMF) has emerged as a crucial bio-based chemical building block in the drive towards developing materials from renewable resources, due to its direct preparation from sugars and its readily diversifiable scaffold. A key obstacle in transitioning to bio-based plastic production lies in meeting the necessary industrial production efficiency, particularly in the cost-effective conversion of HMF to valuable intermediates. Toward addressing the challenge of developing scalable technology for oxidizing crude HMF to more valuable chemicals, here we report coordinated reaction and enzyme engineering to provide a galactose oxidase (GOase) variant with remarkably high activity toward HMF, improved $O_2$ binding and excellent productivity (>1,000,000 TTN). The biocatalyst and reaction conditions presented here for GOase catalysed selective oxidation of HMF to 2,5-diformylfuran offers a productive blueprint for further development, giving hope for the creation of a biocatalytic route to scalable production of furan-based chemical building blocks from sustainable feedstocks.

[1] School of Chemistry, The University of Manchester, Manchester Institute of Biotechnology, Manchester, UK. [2] Department of Chemical and Biochemical Engineering, Technical University of Denmark, Lyngby, Denmark. [3] BASF SE, White Biotechnology Research, Ludwigshafen, Germany. [4] Present address: Novozymes A/S, Krogshoejvej 36, Bagsvaerd, Denmark. ✉email: nicholas.turner@manchester.ac.uk

Biomass waste is an abundant, carbon-rich renewable feedstock that, if processed efficiently, could provide access to many chemicals and fuels as an alternative to those produced from fossil resources. The sugars component of biomass can be chemically dehydrated to give furfural and 5-hydroxymethylfurfural (HMF), of which the latter has been identified by the U.S. Department of Energy as one of the top 12 potential platform chemicals from renewable feedstocks[1]. HMF can be transformed through different reactions to produce a range of derivatives, many having applications in the polymer industry[2–4]. In particular, the oxidized HMF derivatives 2,5-diformylfuran (DFF) and furan-2,5-dicarboxylic acid (FDCA) are both important intermediates in furan based polymer synthesis (Fig. 1), as they can be condensed with other monomers to make poly-imines, -esters, -amides and -urethanes as plastics, resins and porous organic frameworks[2,5,6]. In addition to more environmentally responsible manufacturing methods[7], many of these bio-based plastics have properties that rival those produced from petroleum resources, such as the polyethylene furanoate 'drop in' replacement for polyethylene terephthalate[8,9], and some even display unique properties of potential use in performance materials[10,11].

The primary obstacles preventing the necessary industrial scale production of these furan-based plastics are (1) an inexpensive, continuous and large supply of the basic materials from renewable resources, (2) an efficient and cost effective process to convert them to monomer units, and (3) effective implementation at a production scale that meets the consumer demand. Recent advances in reaction engineering using both inorganic and acid catalysts have enabled high yielding conversions of carbohydrates to HMF[4,12–17]. Yet HMF is unstable, and often needs to be purified to remove salts and by-products left as impurities prior to further chemical conversion, in order to avoid catalyst poisoning[4,14,18]. Once in hand, HMF can be oxidized by a variety of metal catalysts or electrochemical methods to produce DFF, FDCA, or the dimethyl ester of FDCA (furan dicarboxylic methyl ester, FDME) which are more stable monomers[2,4,19,20]. However, these reactions are frequently performed at elevated temperatures and pressures, which can be energy intensive and therefore counterproductive to the aim of developing a sustainable process.

Preferably, biocatalytic oxidation of HMF to these valuable derivatives could extend the sustainability of these bio-based chemicals to include a more environmentally friendly production process. However, this activity is limited to only a few enzymes[21,22], primarily chloroperoxidases[23], HMF oxidases (HMFO)[24], aryl-alcohol oxidases[25,26], and galactose oxidases (GOase) and related copper radical oxidases[27–29]. The latter three seem to be the most appealing due to their dependence on oxygen as a co-substrate rather than $H_2O_2$ as in peroxidases or $NAD(P)^+$ as in dehydrogenases. Nevertheless, this presents a number of issues when working at scale due to limitations in oxygen solubility in aqueous media and thereby supply, that can significantly impact biocatalyst performance[30–34]. HMF oxidases and aryl-alcohol oxidases are also interesting in that they catalyse the complete oxidation of HMF to FDCA[24–26,35–37]. In analogous two-enzyme systems, GOase has been combined with periplasmic aldehyde oxidase PaoABC[28] or unspecific peroxygenase[38,39] to create an oxidative cascade from HMF to FDCA. However, productivity and selectivity of these systems are not yet adequate for large-scale implementation, and none provide access to DFF as the final product and our desired target. In fact, DFF has been particularly challenging to selectively synthesize efficiently (chemically or biocatalytically) without requiring high levels of (bio) catalyst[22,40,41], but GOase and related copper radical oxidases do provide DFF as the main product[27–29].

Building on our previous work in understanding the effect of reaction conditions on GOase for alcohol oxidation (critically, the inclusion of horseradish peroxidase (HRP) to activate GOase and catalase to remove destructive $H_2O_2$)[42], we have engineered GOase for improved kinetic properties for both alcohol oxidation and oxygen binding to deliver a variant that exhibits particularly high catalytic activity for HMF oxidation to selectively produce DFF. Coordinated reaction engineering led to further improvements in conversion and demonstrated that a GOase-based synthesis is amenable to the necessary process intensification required for ultimate industrial implementation. This approach comprising coordinated reaction and enzyme engineering[43,44] has resulted in an effective and high yielding model system for the selective bio-oxidation of HMF to DFF at high substrate loading.

## Results and discussion

For DFF, FDCA and their derivatives to be relevant for use in bio-based plastics, the scale of production needs to meet a market demand, and the production process needs to be cost competitive compared to current methods of producing the non-bio-based monomer equivalents. We began by defining a set of proof of principle reaction metric targets to guide our approach based on estimated threshold values for a biocatalytic processes in the bulk chemical sector[45,46] (Table 1). These metrics were then used to help define which aspects were still in need of improvement before potential future implementation in a true industrial process.

Our approach began with enzyme engineering that was guided by reaction constraints to directly develop a biocatalyst that was industrially relevant, applying screens to select for improvement in turnover rate and for improvements in activity at low oxygen levels. Biocatalyst development was balanced by concurrent, iterative reaction development to tune conditions and push performance limits to identify scalable conditions. This also served as a secondary selection process to identify the top performing variants under potential process conditions. As such, the laboratory scale experiments were performed with the end goal of scale-up in mind. These two approaches proved cooperative, using the results of one to help guide modifications to the other in a so-called 'ideal scenario' of process and biocatalyst driven

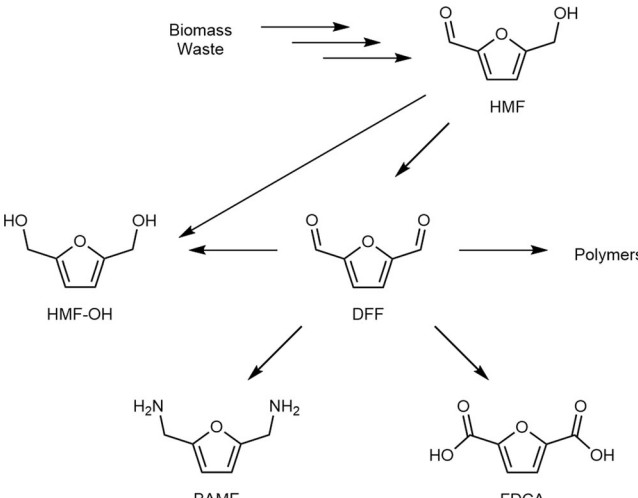

**Fig. 1 Diversification of HMF to high value products via DFF intermediate.** Bio-derived HMF can be chemically and/or enzymatically converted to various other furans that are useful as intermediates or monomers. HMF, 5-(hydroxymethyl)furfural. DFF, 2,5-diformylfuran. HMF-OH, 2,5-bis(hydroxymethyl)furan. BAMF, 2,5-bis(aminomethyl)furan. FDCA, 2,5-furandicarboxylic acid.

**Table 1 Target proof of principle reaction metrics for oxidation of HMF to DFF by GOase.**

| | Unit | Target |
|---|---|---|
| Final Prod. Conc. | g/L | 100 |
| Duration | h | 25 |
| Productivity | g/L.h | 4 |
| Specific Yield[a] | $g_{Product}/g_{Biocatalyst}$[b] | 1000 |
| Conversion | % | 98 |
| Isolated Yield | % | 98 |
| Purity | % | >98 |

[a]Specific yield is defined as (mass product over the biocatalyst lifetime)/(mass biocatalyst).
[b]$g_{Biocatalyst}$ taken as pure enzyme or pure enzyme equivalent in CFE based on specific activity.

process design[44,47]. In the interests of clarity, work on these approaches is discussed in two separate subsections below, despite being coordinated and occurring simultaneously.

**Enzyme engineering**. The first round of library generation targeted pairs of active site residues (Fig. 2), and screening against 1-hexanol identified variant $M_4$ ($M_{3-5}$ + Y329L/M330F mutations) as being the top performing hit (Supplementary Note). Residues outside the active site were also targeted to randomly recombine previously published beneficial mutations into $M_4$. Mutated sites in variants published by Delagrave et al. (variant 7.5.1 with C383S/Y436H/N318D/V477D/A626S/V494A)[48], Wilkinson et al. (variant C383S/Y436H)[49], and Deacon and McPherson (C383S and C383T)[50] were chosen as sites for potential non-selective catalytic improvements. None of these residues interact directly with the substrate (Supplementary Fig. 1), but each variant showed increases in $k_{cat,app}$, reductions in $K_{M,app}$, or both. Mutations other than V494A, which was already present in GOase since the $M_1$ variant (Supplementary Table 1), were randomly introduced to create a library of unique combinations and screened for activity on 1-hexanol. The top hits identified from this screen were variants $M_{5-1}$ (V477D/A626S), and $M_{5-2}$ (N318D/C383T/Y436H/V477D). Interestingly, these hits had distinct sequences compared to the previously published variants. Kinetic characterization of these two variants showed very similar parameters compared to the progenitor (Table 2), therefore, it is possible that the mutations have more influence on GOase expression and/or stability, which contributed to their detection in the screen.

A universal issue in the application of oxygen-dependent enzymes revolves around the balance of providing sufficient oxygen to the bioreactor while minimizing potential enzyme deactivation at the gas-liquid interface and potentially stripping volatile substrate(s), product(s) and/or co-solvent[33,51]. Oxygen has a low solubility in aqueous solution (~270 μM)[31,33,52], while $K_M$ for $O_2$ ($K_{MO}$) for many enzymes is often estimated to be much higher than the oxygen concentration in equilibrium with air[33,53]. This difference means that reaction rates will be dependent upon the oxygen concentration in the liquid and thereby on oxygen transfer rates[42]. This limitation is magnified at high substrate loadings, where substrate concentration and oxygen concentration can differ by a factor of up to $10^3$ [30,32]. The problem results in a system that often prevents the enzyme from operating at maximum performance, and therefore other strategies are required.

One approach to overcome the oxygen problem is to improve the $K_{MO}$ of the biocatalyst, or in other words, engineer the enzyme to work more effectively at lower oxygen concentrations[42,54–57]. Likewise, improvements in $k_{cat}/K_{MO}$ (catalytic efficiency for oxygen) will be more beneficial than improvements in $k_{cat}/K_{MS}$

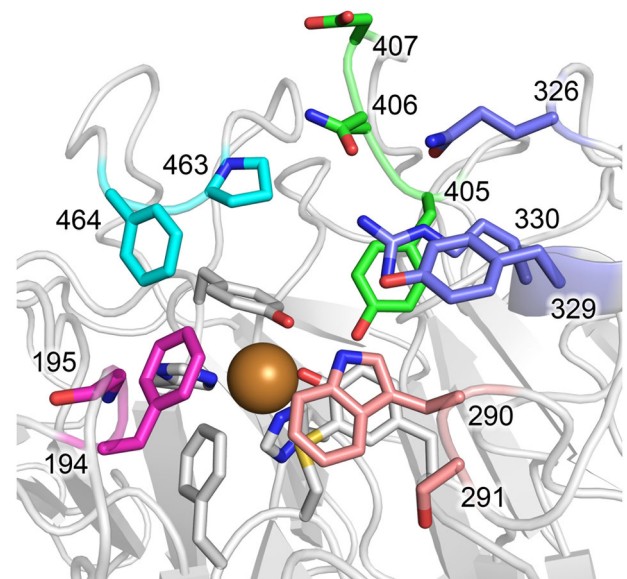

**Fig. 2 Active site residues targeted for mutagenesis.** Positions of active site residues targeted in the seven GOase $M_{3-5}$ NNK CASTing libraries, displayed on wildtype GOase crystal structure (PDBID: 1GOG). Libraries A–G were: A (green): 406/407, B (blue): 326/330, C (pink): 290/291, D (purple): 194/195, E (cyan): 463/464, F (green): 405/406 and G (blue): 329/330.

**Table 2 Apparent kinetic parameters for GOase variants with HMF.**

| Variant Name | $k_{cat,app}$ (s$^{-1}$) | $K_{M,app}$ (mM) | $k_{cat}/K_M$ (s$^{-1}$ M$^{-1}$) |
|---|---|---|---|
| $M_{3-5}$ | 123.9 ± 1.4 | 3.3 ± 0.1 | 38000 ± 1000 |
| $M_4$ | 132.7 ± 2.6 | 2.1 ± 0.2 | 63000 ± 6000 |
| $M_{5-1}$ | 153.9 ± 4.2 | 2.0 ± 0.2 | 77000 ± 8000 |
| $M_{5-2}$ | 139.0 ± 2.4 | 1.8 ± 0.1 | 77000 ± 4000 |
| $M_{6-A}$ | 200.5 ± 13.7 | 10.6 ± 2.8 | 19000 ± 5000 |
| $M_{6-B}$ | 115.4 ± 6.6 | 15.8 ± 2.9 | 7300 ± 1400 |
| $M_{7-1A}$ | 246.2 ± 10.7 | 15.3 ± 2.2 | 16000 ± 2000 |
| $M_{7-2A}$ | 204.6 ± 10.7 | 14.5 ± 2.6 | 14000 ± 3000 |

(where $K_{MS}$ is the $K_M$ for the target substrate). Because the active site of GOase is fairly exposed on the surface of the enzyme, residues nearest to the active site $Cu^{2+}$ ion were expected to have the greatest impact on oxygen binding and reactivity. The solid phase assay proved to be amenable for use within a glovebox, allowing the screen to be performed in a controlled environment containing a low $O_2$ atmosphere (around 0.2% v/v, see Supplementary Note). Using this format, active site libraries C, D, and E (Fig. 2) on the GOase $M_4$ template were screened for improved GOase activity at low oxygen levels using HMF as substrate. Despite the low oxygen concentration, multiple potential hits from libraries C and D were identified. Upon confirmation in a secondary screen and basic characterization in analytical scale biotransformations to challenge these newest variants in current reaction conditions (see below), GOase $M_{6-A}$ (F290W/S291S) was identified as the top performing variant. Additionally, one potential hit from library C (variant $M_{6-B}$, with F290W/S291R mutations) was identified after quickly producing a uniquely large spot on the assay plate following removal after several hours of incubation at the low oxygen atmosphere. The common mutation between these two variants is the reversion of F290 to tryptophan, which is the residue found in the wild-type enzyme and the early GOase $M_1$ variant[58], but was mutated to phenylalanine during engineering of the GOase $M_3$

variant[59] (Supplementary Table 1). It appears that W290 has a significant impact on substrate (HMF and/or $O_2$) binding and catalysis since the $k_{cat,app}$ for HMF in the $M_{6-A}$ variant increased by 50%, while the $K_{M,app}$ increased by approximately 5-fold (Table 2). The $K_{M,app}$ for $M_{6-B}$ was similarly affected, although there was little change in $k_{cat,app}$ indicating that the S291R mutation counteracts the beneficial effect of the F290W mutation in this variant.

The final round of enzyme engineering was based on merging the results of the two evolution strategies to create the $M_{7-1A}$, $M_{7-2A}$, $M_{7-1B}$, and $M_{7-2B}$ variants (Supplementary Table 1). $M_{7-1A}$ and $M_{7-2A}$ were kinetically characterized, revealing an increase in $k_{cat,app}$ to 246 s$^{-1}$ for $M_{7-1A}$ (Table 2), which compares quite favourably to FAD-dependent enzymes that oxidize HMF[24–26,36,37], and is similar to another copper radical oxidase variant[27]. In this instance, the combination of mutations from $M_{5-1}$ and $M_{5-2}$ with $M_{6-A}$ appears to be additive, which is a reflection of the non-specific catalytic enhancement provided by such distal mutations.

The kinetic measurements above are apparent values since they were determined under oxygen limited conditions. A more complete dataset was collected for selected GOase variants using the recently developed Tube-in-Tube Reactor (TiTR)[60], which has enabled a simplified method for accurately determining true kinetic parameters for oxygen-dependent enzymes, specifically $K_{MO}$[33]. The variants chosen for these detailed characterizations were selected to sample the improvements over the course of the engineering program in comparison to the wild-type ($M_1$) variant: the initial $M_{3-5}$ progenitor, the first hit for improved HMF activity ($M_4$) and the top hit from the low $O_2$ screen ($M_{6-A}$).

$K_{MO}$ values changed significantly over the course of evolution, increasing drastically in early variants before returning to wild-type levels after screening in the low oxygen environment (Table 3). The influence of $K_{MO}$ can be observed by comparison of the TiTR data with the apparent kinetic parameters reported in Table 2. Measurements for the $M_4$ and $M_{6-A}$ variants are fairly consistent between the two experiments as a result of $K_{MO}$ values near or below the level of aqueous $O_2$ concentration in equilibrium with air (~270 µM and 30 °C). However, the high $K_{MO}$ for GOase $M_{3-5}$ shows that this variant was severely oxygen limited in the standard kinetic assay and was therefore much more responsive to the increase in aqueous $O_2$ concentration in the TiTR. The improvement in $k_{cat}/K_{MO}$ found in the $M_{6-A}$ variant as a result of the low oxygen assay conditions highlights the value of using this screen for engineering oxygen dependent enzymes, and confirms that it is indeed possible to intentionally target and improve reactivity in GOase. Furthermore, we believe that this glovebox modification could be generally applied to provide a unique selective pressure for engineering $K_{MO}$ in other oxygen-dependent enzymes, which could significantly accelerate their development for use at industrial scale.

**Reaction engineering**. Reaction engineering for the GOase catalysed oxidation of HMF to DFF was performed alongside the enzyme engineering work, incorporating newly improved variants

and substrate formulations as they became available. These reactions were also used as a secondary assay to evaluate performance of hits from library screens to identify best performing variant of the round of evolution. Initial reaction compositions and conditions were based on our previously published work[42] and were then adapted in stages to target the effects of individual parameters. As described below, this approach allowed an assessment of the limits for a given variant while also considering requirements of future process implementation.

Initial reactions to characterise over-oxidation of DFF to 5-formyl-2-furan carboxylic acid (FFCA) by GOase variants[61] indicated that this was not a concern, and identified $M_4$ as the more effective biocatalyst for HMF oxidation (Supplementary Note and Supplementary Table 2). From there, we next started a process of reaction optimization beginning with screening for compatible co-solvents. An initial screen of typical solvents highlighted DMSO as a beneficial and compatible solvent, and 0.05 g/L (0.71 µM) purified GOase as an ideal working biocatalyst loading giving high conversion (94%) of 100 mM HMF and good conversion (71%) of 250 mM HMF (Supplementary Note and Supplementary Tables 3 and 4). Additionally, GOase $M_4$ performed well when challenged with a preparation of crude HMF from BASF (71% conversion of 25 g/L, Supplementary Table 5). An important consideration in the accessibility of furan-based materials from biomass is the processing necessary to convert the crude HMF into a formulation suitable for the subsequent reaction. As mentioned, HMF is frequently required in the purified form before it can be converted satisfactorily by many inorganic catalysts, thus the option to overcome such purification requirements by using crude HMF could lead to significantly reduced production costs. Similarly, use of a crude biocatalyst formulation such as dried cell free extract could prove more cost-effective. In the first instance, an approximately equivalent GOase $M_4$ loading of dried CFE formulation (5–6% GOase by weight) gave slightly higher conversions than the purified enzyme (66% conversion compared to 50% at 250 mM, Supplementary Table 6).

Attempting to remove DMSO to avoid future process complications resulted in reduced productivity (comparing Supplementary Table 4 Entry 3 and Supplementary Table 6 Entry 1). To find a more suitable solvent for scale-up, we then examined $M_4$ performance in a variety of water-miscible and -immiscible co-solvents (Supplementary Tables 7 and 8), in addition to the effect of reaction temperature (Supplementary Table 9). The most promising results were observed when using ethyl acetate (EtOAc) at 20 °C. It is expected that both the addition of a water-immiscible organic solvent and the lower temperature contribute to improved biocatalyst stability by reducing the aqueous concentration of the dialdehyde product. Dialdehydes are common crosslinking reagents[62] and can modify accessible lysine residues (22 in the case of this construct), often reducing enzyme activity and stability. Accordingly, effective removal of the DFF product should enable higher conversions.

**Table 3 Kinetic parameters of GOase variants determined in the TiTR with HMF.**

| Variant | $K_{MO}$ (mM) | $K_{MS}$ (mM) | $k_{cat}$ (s$^{-1}$) | $k_{cat}/K_{MO}$ (M$^{-1}$ s$^{-1}$) | $k_{cat}/K_{MS}$ (M$^{-1}$ s$^{-1}$) |
|---|---|---|---|---|---|
| $M_1$ | 0.16 ± 0.11 | 53 ± 18 | 23.5 ± 4.7 | $(1.5 ± 1.1) \times 10^5$ | $(0.04 ± 0.02) \times 10^4$ |
| $M_{3-5}$[a] | 1.39 ± 0.46 | 14.9 ± 4.5 | 651 ± 132 | $(4.7 ± 1.8) \times 10^5$ | $(4.4 ± 1.6) \times 10^4$ |
| $M_4$ | 0.37 ± 0.07 | 1.83 ± 0.30 | 120 ± 6 | $(3.3 ± 0.6) \times 10^5$ | $(6.6 ± 1.1) \times 10^4$ |
| $M_{6-A}$ | 0.15 ± 0.02 | 6.19 ± 0.41 | 166 ± 4 | $(10.9 ± 1.5) \times 10^5$ | $(2.7 ± 0.2) \times 10^4$ |

$K_{MO}$: $K_M$ for oxygen, $K_{MS}$: $K_M$ for HMF. [a]Measured using CFE powder with an estimated GOase content, giving a higher uncertainty in the $k_{cat}$ parameter. Data previously published in the PhD thesis of A. T. P.[77].

**Table 4 Conversion of HMF to DFF by GOase variant M$_4$ CFE using EtOAc overlay at reduced incubation temperature.**

| Entry | GOase Variant | GOase (g/L) | GOase Form | HRP (g/L) | [HMF] | Type HMF | Conversion (%) | Over-oxidation (%) |
|---|---|---|---|---|---|---|---|---|
| 1 | M$_4$ | 0.625 | CFE | 0.0128 | 250 mM | Pure | 96 | 2.5 |
| 2 | M$_4$ | 0.625 | CFE | 0.0128 | 25 g/L | Crude | 88 | 6.4 |
| 3 | M$_4$ | 0.625 | CFE | 0.0128 | 500 mM | Pure | 71 | 0.4 |
| 4 | M$_4$ | 0.625 | CFE | 0.0128 | 50 g/L | Crude | 82 | 0.8 |

Data for reactions with purified enzyme provided in Supplementary Table 10. Conditions listed in the table highlight the main differences between samples, while those listed below the table apply to all entries unless otherwise stated.
Conditions: 0.05 mM CuSO$_4$ in GOase M$_4$ CFE reactions, 880 U/mL Catalase, 100 mM NaPi, 200 μL (80% of aqueous volume) EtOAc, 20 °C, 6 h.

**Table 5 Conversion of HMF to DFF by GOase variant M$_4$ CFE at high HMF loading in the presence of different cosolvents.**

| Entry | GOase Variant | GOase (g/L) | GOase Form | HRP (g/L) | Co-Solvent | [HMF] (g/L) | Type HMF | Conversion (%) | Over-oxidation (%) |
|---|---|---|---|---|---|---|---|---|---|
| 1 | M$_4$ | 0.625 | CFE | 0.0128 | EtOAc | 50 | Crude | 82 | 0.8 |
| 2 | M$_4$ | 0.625 | CFE | 0.0128 | DEC | 50 | Crude | 82 | 0.8 |
| 3 | M$_4$ | 0.625 | CFE | 0.0128 | BuOAc | 50 | Crude | 76 | 0.7 |

Data for reactions with purified enzyme provided in Supplementary Table 11. Conditions listed in the table highlight the main differences between samples, while those listed below the table apply to all entries unless otherwise stated.
Conditions: 0.05 mM CuSO$_4$ in GOase M$_4$ CFE reactions, 880 U/mL Catalase, 100 mM NaPi, 200 μL (80% of aqueous volume) solvent overlay, 20 °C, 6 h.

Integrating these two beneficial reaction features (i.e. 20 °C and EtOAc layer) gave an improved set of performance metrics: almost complete conversion of 250 mM HMF and 25 g/L crude HMF by GOase M$_4$ as purified enzyme and as CFE after 6 h, suggesting that substrate loading could be pushed still higher (Table 4 and Supplementary Table 10). Indeed, under the same conditions, conversion to DFF reached 71% of 500 mM HMF and 82% of 50 g/L crude HMF by M$_4$ CFE after 6 h.

Large scale application of GOase will likely require a continuous supply of air (or oxygen enriched air) to meet the oxygen requirements for substrate conversion. Consequently, low boiling solvents, such as EtOAc (bp 77 °C), would be stripped out of the reaction vessel, even with the installation of an effective condenser. We therefore evaluated organic solvents with lower vapour pressures but with log P values similar to EtOAc (Supplementary Note, Supplementary Fig. 2). Co-solvents that proved to be more beneficial and compatible with GOase were esters, carbonates and ethers, with diethyl carbonate (DEC, bp 126°) and butyl acetate (BuOAc, bp 126 °C) identified as the most promising (for unfavourable solvents, see Supplementary Note). DEC was found to give comparable levels of conversion at both 250 and 500 mM HMF using purified GOase M$_4$, and also at 50 g/L crude HMF when using GOase M$_4$ as CFE (Table 5 and Supplementary Table 11). This high level of conversion of crude HMF by the unpurified GOase preparation as CFE (Table 5 Entry 2) now approaches industrially relevant metrics (Table 1), while utilizing a biocatalyst formulation that is more realistic for industrial application. Working at these high substrate loadings, we suspected that the benefits of the biphasic reaction were two-fold. In effect, the second liquid phase was acting as a reservoir both for HMF as well as DFF, continuously feeding the substrate while extracting the product. These expectations are generally supported by the experimentally determined partition of HMF and DFF between the aqueous and organic layers (Supplementary Table 12).

The four newly identified Round 2 GOase variants displayed greatly improved conversion of 50 g/L semi-crude HMF compared to the M$_4$ variant (see Supplementary Note and Supplementary Table 13), with the best variant (M$_{6-A}$) reaching nearly 90% conversion after 6 h. This was particularly significant

**Table 6 Conversion of 100 g/L semi-crude HMF to DFF by GOase variants.**

| Entry | GOase Variant | Conversion at 6 h (%) | Conversion at 24 h (%) |
|---|---|---|---|
| 1/2 | M$_4$ | 31 | 31 |
| 3/4 | M$_{5-1}$ | 38 | 39 |
| 5/6 | M$_{5-2}$ | 51 | 54 |
| 7/8 | M$_{6-A}$ | 57 | 62 |
| 9/10 | M$_{6-B}$ | 59 | 76 |
| 11/12 | M$_{7-1A}$ | 64 | 69 |
| 13/14 | M$_{7-2A}$ | 77 | 77 |
| 15/16 | M$_{7-1B}$ | 64 | 81 |
| 17/18 | M$_{7-2B}$ | 58 | 78 |

Conditions: 0.05 g/L pure enzyme, 0.0064 g/L HRP, 880 U/mL Catalase, 100 mM NaPi, 100 g/L Crude HMF in DEC (BASF), 200 μL (80% of aqueous volume) DEC, 20 °C, 6 and 24 h. Over-oxidation in all samples was found to be <0.3%.

since the variants were engineered in two distinctly different screening strategies: in one case targeting distal residues for increased k$_{cat}$, and in the other targeting active site residues for lower K$_{MO}$. The combination of these mutations in Round 3 variants M$_{7-1A}$, M$_{7-2A}$, M$_{7-1B}$ and M$_{7-2B}$ led to an additive effect and still greater conversion than the parental variants (up to 81%) when tested at 100 g/L semi-crude HMF (Table 6). All variants holding the unique M$_{6-A}$ 'A' mutations (F290W/S291S) had a faster rate of conversion and high overall conversion, which reflects the increased k$_{cat,app}$ values measured for these variants (Table 2). Additionally, variants having the unique M$_{6-B}$ 'B' mutations (F290W/S291R), were the only variants that showed a significant increase in conversion after the 6 h time point, suggesting that the additional S291R has a beneficial effect on stability, potentially via formation of a salt bridge with E195. However, the S291R mutation appears to prevent a similar improvement in turnover rate provided by F290W in the M$_{6-A}$ variant.

With an interest to identify a broader range of solvents that could potentially improve downstream processing of the reaction,

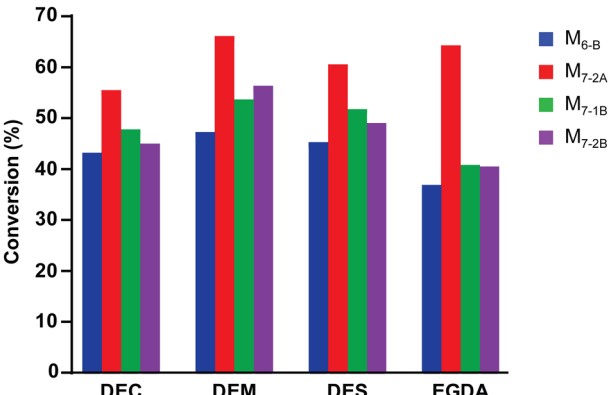

**Fig. 3 Solvent screening for improved conversion at high HMF loading.** GOase variants were evaluated for conversion of 150 g/L semi-crude HMF to DFF in the presence of different cosolvents. Conditions: 0.05 g/L pure enzyme, 0.0064 g/L HRP, 880 U/mL Catalase, 100 mM NaPi, 150 g/L Crude HMF in solvent (BASF), 200 μL (80% of aqueous volume) solvent overlay, 20 °C, 24 h. No over-oxidation was observed in any of the samples. DEC, diethyl carbonate. DEM, diethyl malonate. DES, diethyl succinate. EGDA, ethylene glycol diacetate. Each bar represents a single experiment.

several other co-solvents that offered slightly different physical properties (density, Log P, water solubility, boiling point, etc., Supplementary Table 14) were identified for compatibility testing with GOase. Of these solvents, diethyl malonate, diethyl succinate, and ethylene glycol diacetate (DEM, DES and EGDA, respectively) gave similar conversions as the standard reaction with DEC at 100 g/L semi-crude HMF, while all others gave significantly lower conversion (Supplementary Fig. 3). These four solvents (DEC, DEM, DES and EGDA) were then used to evaluate the top performing GOase variants ($M_{6-B}$, $M_{7-2A}$, $M_{7-1B}$ and $M_{7-2B}$) challenged with 150 g/L (~1.2 M) semi-crude HMF (Fig. 3). Despite the apparent stability improvements of the 'B' variants, GOase $M_{7-2A}$ was consistently the most productive in all reaction compositions, reaching up to 66% conversion after 24 h with DEM as the co-solvent, remarkably approaching 100 g/L product concentration.

A calculation of total turnover number under these conditions with 0.05 g/L (0.71 μM) purified GOase translates to an exceptionally high total turnover number (TTN) of $1.11 \times 10^6$ for the GOase $M_{7-2A}$ variant in DEM, giving an excellent comparison to TTN ranges for other industrially applicable biocatalysts[63–67]. The highest productivity though was achieved with GOase $M_{7-2A}$ at 100 g/L semi-crude HMF in DEC after 6 h (Table 6 Entry 13), reaching 12.8 g/L.h (aqueous volume only, or 7.1 g/L.h when including solvent volume) and a specific yield 1500 $g_{DFF}/g_{enzyme}$. This represents a nine-fold increase in productivity with a seven-fold increase in specific yield when compared to the $M_{3–5}$ progenitor enzyme at 1.4 g/L.h and 218 $g_{DFF}/g_{enzyme}$ (Supplementary Table 2 Entry 7).

In an effort to increase HMF conversion at high substrate loadings (100 and 150 g/L semi-crude HMF in DEC), reactions were performed with 0.1 g/L GOase $M_{7-2A}$. After 6 h, 100 g/L HMF was almost fully converted to DFF (96%), while conversion of 150 g/L loading lagged behind (62%) (Supplementary Table 15). In both cases extended reaction times led to little further increase in conversion.

**Preliminary reaction testing**. Relatively successful preliminary 10 mL scale reactions during early stage reaction engineering provided a benchmark against which to measure reactions at larger scale (Supplementary Table 16). GOase variants $M_{3–5}$, $M_{6-A}$

and $M_{7-2A}$ were selected to highlight the improvements in performance gained through the low oxygen screening conditions ($M_{6-A}$) and recombined best mutations ($M_{7-2A}$) compared to the initial progenitor ($M_{3–5}$).

Conditions for the 0.2 L reactions were evaluated to set initial parameters to allow comparison of the variants (Supplementary Note). After 6 h, GOase $M_{3–5}$ reached 24% conversion of 50 g/L HMF, while GOase $M_{6-A}$ and $M_{7-2A}$ had similarly improved performance at 37 and 36% conversion, respectively (Supplementary Fig. 4a and Supplementary Table 17 Entries 1–3). This conversion was lower than expected from the analytical scale reactions, which we surmised was at least partly due to the very low (~50 U/mL) catalase loading compared to the analytical scale reactions (440 or 880 U/mL) not being able to convert $H_2O_2$[42]. Indeed, when the catalase concentration was increased in reactions using $M_{7-2A}$, up to 59% conversion was observed after 6 h (Supplementary Fig. 4b and Supplementary Table 17 Entries 4–5) indicating that catalase activity was a limiting factor. These reactions, however, also exhibited some foaming. Additional complications were observed in the transition to this scale, with a significant amount of DEC lost due to stripping in the bubbled reactor, and the formation of emulsions, both of which would likely influence biocatalyst performance.

By comparison to the best analytical scale reactions, the 0.2 L reaction at high catalase loading gave a volumetric productivity of 4.8 g/L.h (aqueous volume only, or 2.6 g/L.h when including solvent volume) and specific yield of 27.3 $g_{DFF}/g_{biocatalyst}$ as CFE, or 516 $g_{DFF}/g_{biocatalyst}$ as pure enzyme equivalent (Supplementary Table 17). As an additional separate validation at scale, a version of this biphasic reaction was performed by BASF at a total volume of 1.44 L (0.8 L aqueous, 0.64 L DEC) using 0.05 g/L GOase with 31.5 g/L HMF loading. Here, the reaction resulted in 92% isolated yield of DFF (at a specific yield of 570 $g_{DFF}/g_{biocatalyst}$), demonstrating the feasibility of this reaction design.

The results presented here highlight the value of harmonizing informed reaction and process considerations into the enzyme engineering workflow to directly develop a biocatalyst that balances process and enzyme requirements. Using this coordinated approach to reaction and enzyme engineering, we have developed an advanced GOase variant and corresponding reaction conditions for effective production of the bio-based chemical building block DFF that could be used directly in further enzymatic (or chemo-catalytic) transformations[28,68,69] to produce other key bio-derived furans for industrial applications. The low catalyst loading of our GOase $M_{7-2A}$ variant combined with the achievement of high conversion and TTN at >1 M substrate concentrations is unique for biocatalysts active on HMF, and is additionally distinctive in its selective oxidation to the dialdehyde product. To the best of our knowledge, this engineered GOase and the associated reaction conditions represent the most efficient and selective biocatalytic production of DFF to date, and certainly compares favourably with available chemo-catalytic synthetic methods[22,40,41].

Despite these advances, there are still several aspects that need further work to reach and surpass the performance metrics outlined in Table 1, in addition to a variety of considerations for translation to industrial scale. For instance, further improvements in enzyme stability are necessary to enable high conversion at increased substrate loadings to elevate production titres to meet expected production scale demands. These could potentially be found through additional enzyme engineering to improve aldehyde tolerance and biocatalyst lifetime, for example, but ultimately process development at a scale that would allow in situ product removal (ISPR)[70] is likely required to bring productivity and specific yield into a more viable range. Additionally, replacement of HRP as GOase activator with a chemical[71] or

electrochemical[72] oxidant may provide a cheaper and equally effective alternative that can be used as a 'drop-in' substitute within this system. Furthermore, methods for bulk biocatalyst production in an expression host need to be re-validated for the GOase variant presented here, presumably based on the well-established *Pichia* expression systems for GOase[73]. Biocatalyst production in this way may also, at least partially, alleviate foaming issues due to a purer biocatalyst formulation, as was observed in the validation reaction by BASF. Finally, identifying a continuously available carbohydrate feedstock to supply an inexpensive and reliable method for HMF synthesis[4,12–17] will be critically important to allow uninterrupted production of DFF for use in industry. Although these obstacles still stand in the way of commercial production of DFF, each has significant precedent to be achievable and the largest truly missing piece was a highly productive (bio)catalyst for HMF oxidation. With this prototype biocatalyst now in hand, broader production and use of these bio-based furan polymers is closer to being in sight.

## Methods

**Materials**. All reagents used were of the highest grade available from Sigma-Aldrich or Fisher Scientific and used without further purification. Libraries on the GOase $M_{3-5}$ template in pET30 were synthesized by DNA2.0 to have NNK degeneration at codons corresponding to the previously published GOase libraries A, B, C, D, E, F and G[74]. Mutagenic primers were purchased from MWG Eurofins. Cell free extract (CFE) powders were produced by Prozomix Ltd unless otherwise stated. A codon optimized version of GOase $M_{7-2A}$ was used for production of $M_{7-2A}$ CFE for the scale up biotransformation due to poor expression of this variant. Our standard GOase $M_{3-5}$ and GOase $M_{6-A}$ constructs were used for the other preparations. The gene sequence was optimized by IDT, synthesized by LifeTechnologies and cloned into pET30 at NheI/XhoI sites using NEBuilder according to the manufacturer's protocol.

**Galactose oxidase expression and purification**. Galactose oxidase variants were expressed and purified as previously described[75]. The pET30 construct of GOase variant was transformed into BL21 Star (DE3) cells (Invitrogen) according to manufacturer's protocol and grown overnight at 37 °C on LB/agar plate containing 30 μg/mL kanamycin. A single colony was picked to inoculate an overnight culture of 5 mL LB supplemented with 30 μg/mL kanamycin and grown at 37 °C. 500 μL of this overnight culture was used to inoculate 250 mL of 8ZY-4LAC-SUC auto-induction media[50] and grown at 26 °C for approximately 64 h. Cells were harvested by centrifugation at $2800 \times g$ at 4 °C for 20 min. The cell pellet was resuspended in 25 mL lysis buffer (50 mM PIPES, 25% sucrose (w/v), 5 mM $MnCl_2$, 1% Triton X-100 (v/v), and 1 mg/mL lysozyme) and gently shaken at 4 °C for 20–30 min before being lysed by sonication (20 s on, 20 s off, 20 cycles). The clarified lysate (centrifuged $39000 \times g$ for 30 min at 4 °C) was transferred to 30 kDa cut-off dialysis tubing and dialyzed into NP buffer (50 mM NaPi, 300 mM NaCl, pH 8.0) overnight with stirring at 4 °C. The dialyzed sample was passed through a 0.45 μm syringe filter and applied to one or two (in series) 5 mL Strep-Tactin Superflow Plus columns (equilibrated with NP buffer) using a peristaltic pump. The column flow through was collected and applied to the column a second time. After loading the crude lysate, the column was washed with 5 CV of NP buffer followed by elution with 70 mL of NPD buffer (NP buffer with 5 mM desthiobiotin) collecting ~5 mL fractions. Fractions containing GOase (as determined by SDS gel) were combined and concentrated using Sartorius Vivaspin 20 (30 kDa cut-off) and then dialyzed (30 kDa cut-off tubing) overnight at 4 °C in 50 mM NaPi pH 7.4 saturated with $CuSO_4$ to copper load the enzyme. Excess copper was removed by dialysis in 50 mM NaPi pH 7.4 overnight at 4 °C and protein samples were concentrated by Vivaspin before being aliquoted and frozen in liquid $N_2$ prior to storage at −80 °C.

**Mutagenesis methods**. Site-directed mutagenesis was performed following the manufacturer's protocol from the QuikChange II Site-Directed Mutagenesis kit using Phusion (NEB) polymerase. Multisite-directed mutagenesis was performed using the QuikChange Multi Site-Directed Mutagenesis kit (Agilent) according to the manufacturer's protocol. Primers used in the protocols are listed in Supplementary Table 19.

**Solid phase screen on benchtop**. The solid phase GOase library screen was performed similarly to that previously published[76]. The library plasmid pool was transformed into chemically competent BL21Star(DE3) *E. coli* and spread onto a nylon membrane (Roche, 132 mm) on LB/agar containing 50 μg/mL kanamycin. After growth at 26 °C for 40 h, the library was replicated onto a new membrane and grown at 37 °C overnight on a fresh LB/agar/kanamycin plate while the original membrane was used for the assay. Colonies on the membrane were permeabilized by incubation in a chloroform vapour chamber for 10 min at room temperature.

Afterward, a 10 mL solution of reaction mix containing 100 mM 1-hexanol, 0.8 mM $CuSO_4$, 0.1 g/L HRP and 4 mM 4-chloronaphthol in 100 mM NaPi, pH 7.0 with 20% DMSO was combined with 10 mL 2% agarose solution and poured over the membrane to begin the assay. Approximately 3000 colonies per library were screened, with colonies that turned dark purple indicating activity on 1-hexanol. Positive clones were picked from the corresponding replicated membrane the next day, and grown overnight in 5 mL LB supplemented with kanamycin. To confirm hits, a small aliquot of the overnight culture was diluted in LB, re-streaked onto nylon membranes (Roche, 82 mm) on LB/agar containing 50 μg/mL kanamycin and grown at 26 °C for 40 h. Colonies were permeabilized and assayed as described above combining 5 mL of both reaction mix and agarose solutions to pour over the membrane. DNA from the top hits from the screen (i.e. those changing colour the fastest) was collected via MiniPrep (Qiagen) from the remainder of the 5 mL overnight culture and submitted for sequencing to identify mutations.

This assay was also performed with 25 mM 1-hexanol alcohol to screen approximately 1000 colonies of the multisite mutagenesis library created on GOase $M_4$ for each substrate.

**Solid phase screen in glovebox**. To screen for GOase variants with improved activity at low oxygen levels, the solid phase assay was performed as described above for libraries C, D, and E made on the GOase $M_4$ template with a few modifications. Plastic ware, glassware, water bath, chloroform, water, 100 mM NaPi pH 7.0 and 50 mM $CuSO_4$ were placed into a glovebox and purged with nitrogen until the oxygen sensor read ~0.2% $O_2$ the evening before performing the assay. Bottles and tubes of all liquids were open within the glovebox to allow dissolved oxygen levels to equilibrate overnight. Stock solutions of HRP, 4-chloronaphthol and HMF were made and equilibrated in the glovebox before the assay. Low-oxygen water from the glovebox was used to dissolve agarose to make the 2% solution, which was then returned to the glovebox, aliquoted and kept in a 55 °C water bath until needed. After replicating the membrane, the original membrane was transferred to the glovebox to permeabilize the colonies and begin the assay as described above using the components equilibrated within the glovebox with 50 mM HMF as the substrate. In addition to the positive clones identified at low $O_2$, the speed at which colonies changed colour after removal from the glovebox was also noted to identify further hits as necessary. As above, clones were picked from the replicated membrane the next day, grown in LB and re-streaked onto a membrane to validate the hits in a secondary assay at low $O_2$. DNA from the top-performing hits was isolated and sequenced to identify mutations.

**Kinetics**. Kinetic measurements were performed for purified GOase variants using an ABTS/HRP coupled assay as previously described[42]. Activity was measured on a Tecan Infinite M200 plate reader (Magellan V6.5 software) in triplicate at 30 °C. Reactions in a 96-well plate contained 10 μL GOase dilution in 90 μL reaction mix (containing 0.23 g/L HRP and 0.4 g/L ABTS in 100 mM sodium phosphate buffer (NaPi) at pH 7.4). The reaction was initiated by addition of 100 μL substrate solution in water with 20% DMSO. Initial rates were normalized by GOase concentration before calculating kinetic parameters using GraphPad Prism 7. Final HMF concentrations in the 200 μL assay were 0–25 mM or 0–100 mM.

**Tube-in-Tube assay for $k_{mo}$ determination**. Reaction components were introduced into the TiTR as four separate stock solutions using individual syringe pumps. The stocks contained A) buffer, catalase, $CuSO_4$; B) buffer, catalase, $CuSO_4$, HMF; C) buffer, GOase; D) buffer, HRP. It was important for HRP and GOase to remain separate until beginning the reaction as a premixed solution lead to significant deactivation of the GOase over the course of the experiment. Stock solutions were made in 100 mM NaPi, pH 7.4. Catalase and HRP concentrations corresponded to an activity of 20 U/mL and 1 U/mL, respectively, in the reactor. $CuSO_4$ in stock solutions corresponded to a concentration of 100 μM in the reactor. GOase concentration was varied depending on activity of the variant tested to provide and initial rate between 0.2–0.5 mmol/L/min.

Pumps were primed with the appropriate stock solution, and then set to run continuously to reach a steady state. The mass-flow controllers were set to produce a defined mixture of oxygen and nitrogen, and the system was set to wait a total of three residence times for all set-points to ensure steady state had been reached. After this, the sample was injected, and the ramp method was started. The ramp method gradually lowers flow rate from the initial value to increase the residence time, without changing the inlet concentration of any component. Analysis of the reactor output provides precise time series data for the experiment. The slope of the residence time ramp, α, was set to 0.5 for all experiments reported here, and the ramp in flow rate was run for 15 min (real time) to provide data from a residence time of 2 min to 10 min. Samples were collected every minute to provide 11 samples for each experiment measuring initial rate. The percentage of oxygen in the gas and the flow rate for each syringe pump were all controlled by LabVIEW, which automatically adjusted the set-point according to a predefined list of set-points for the experiment.

Samples from the TiTR were analysed on the diode array UV/vis detector. Before injection, the detector was balanced and zeroed. Once the system stabilized and reported as ready, the injection valve was turned to inject the sample via the flowing mobile phase. Spectral data was collected over time (20 Hz frequency) to

collect 3-dimensional data of time-wavelength-absorbance. Absorbance was measured from 210 to 600 nm with a slit width of 4 nm and a step width of 1 nm.

The four stock solutions were made as listed above for characterisation of the GOase variants. The solution containing the GOase variant was pumped at a constant ratio of 1/5 of the total flow rate through the reactor, while the flow ratio of buffer and HMF solutions were varied according to the HMF concentration set-point. Gas flow was kept constant at 1 NL/min, while the volume fraction of oxygen was varied from 5–100%. The reactor was pressurized to 6 bar to provide sufficiently high oxygen concentrations in the reaction solution.

Data handling and parameter estimation was performed using MATLAB (MathWorks, Natick, MA, USA). Raw data from LabVIEW were converted to concentrations of HMF and DFF using a calibration model. A linear model was fitted to the concentration-time data to determine initial rate as slopes, and the data was discarded if the model did not describe the data sufficiently well ($R^2 < 0.9$). Parameters of the enzyme kinetic model (ping pong bi bi) were estimated using non-linear least square regression and including the uncertainty of the initial rate data. Because the UV absorbance spectra of HMF and DFF overlap, multivariate data analysis was needed to determine the substrate and product concentrations. To do this, a chemometric model was made and calibrated using the PLS_Toolbox (Eigenvector Research, Manson, WA, USA) for MATLAB. Calibration curves using for HMF and DFF using this model are shown in Supplementary Fig. 5.

**Analytical Scale biotransformation and HPLC analysis**. The general analytical scale biotransformation (250 μL) contained 0.05 g/L purified GOase variant, 1.7 U/mL (0.0064 g/L) HRP, 880 U/mL (0.08 g/L) catalase, and 250 mM HMF in 100 mM NaPi pH 7.4. Deviations from this are described in the text and include changes in enzyme loading, enzyme formulation, miscible or immiscible cosolvent, substrate loading, substrate purity, buffer strength or the addition of $CuSO_4$. After addition of HMF to start the reaction, tubes were shaken at 250 rpm at the temperatures described, however 20 °C was typically used. Reactions were opened every hour (up to 8 h) to refresh oxygen in the headspace if no time point sample was needed at the given duration.

Aliquots (10 μL) from reactions of up to 100 mM HMF with water miscible solvents were quenched by addition to 75 μL acetonitrile and 10 μL 0.5 M $H_2SO_4$. Samples were diluted to 500 μL with $H_2O$, then centrifuged and transferred to a filter vial before HPLC analysis. For reactions with higher HMF loadings, a 10 μL aliquot was quenched in 225 μL acetonitrile and 10 μL 0.5 M $H_2SO_4$ then diluted to 1500 μL before analysis.

Reactions with water immiscible solvents were quenched by addition of 100 μL 0.5 M $H_2SO_4$ and 150 μL of the corresponding solvent. The sample was vortexed, centrifuged, and the organic layer was removed to dry over $Na_2SO_4$. The aqueous fraction was extracted twice with 750 μL ethyl acetate, combining all organic layers on $Na_2SO_4$. From the aqueous fraction, a 10 μL aliquot was combined with 75 μL acetonitrile and 415 μL $H_2O$ and then centrifuged and transferred to a filter vial for analysis. A 250 or 500 μL aliquot of the organic layer (lower volume for higher HMF concentration) was dried by GeneVac, and the resulting solid was dissolved in 750 μL acetonitrile. The sample was prepared for analysis by combining 37.5 μL of this with 37.5 μL acetonitrile, 5 μL 0.5 M $H_2SO_4$ and 420 μL $H_2O$, centrifuged and transferred to a filter vial for analysis.

Analysis was performed on RP-HPLC (Agilent 1200 Series, ChemStation software or Agilent 1260 Infinity II, OpenLAB CDS software) by injection of 10 μL onto Bio-Rad Aminex HPX-87H column. An isocratic method for 35 min with 15% acetonitrile / 85% 5 mM $H_2SO_4$ at 0.5 mL/min and 60 °C was used to separate the analytes. Absorbance at 280 nm was used for analysis after normalizing signals to compound response factor at equal concentrations or through comparison to a standard curve based on peak area. Crude HMF samples are analysed using a 50 min method with the same conditions. Retention times are 23.0 min for HMF, 26.9 min for DFF, 16.2 min for FFCA and 12.1 min for FDCA. Supplementary Fig. 6 shows the calibration curves for HMF, DFF, FFCA and FDCA.

**Analytical scale reaction for high boiling point solvents**. Analytical scale biotransformations with higher boiling point solvents were performed similar to as described above. The final 250 μL reactions contained 0.05 g/L purified GOase variant, 1.7 U/mL (0.0064 g/L) HRP, 880 U/mL (0.08 g/L) catalase in 100 mM NaPi pH 7.4 with 100 or 150 g/L (aqueous concentration) semi-crude HMF from BASF in the appropriate solvent (200 μL). Semi-crude HMF from BASF was provided in diethyl carbonate. The diethyl carbonate 500 μL aliquots of this preparation was removed by GeneVac, and the residue was dissolved in either propylene carbonate, diethyl oxalate, diethyl malonate, diethyl succinate, ethylene glycol diethyl ether, or ethylene glycol diacetate to give 500 μL stock solutions of the same concentration as the original in diethyl carbonate. HMF in the respective solvent was added to start each reaction, which were then shaken at 250 rpm at 20 °C. Reactions were opened every hour (hours 1–6) to refresh the oxygen in the headspace.

Reactions were quenched by addition of 100 μL 0.5 M $H_2SO_4$ and 150 μL of the corresponding solvent. The sample was vortexed, centrifuged, and the organic layer was removed to dry over $Na_2SO_4$. The aqueous fraction was extracted twice with 750 μL ethyl acetate, combining all organic layers on $Na_2SO_4$. A sample of the dried organic layer was diluted 20-fold in ethyl acetate prior to analysis by GC-FID (Agilent 6850 GC, ChemStation software, Agilent HP-1, 30 m × 0.32 mm × 0.2 μm column) using one of two methods:

Method 1: Hold 100 °C for 4 min, ramp 25 °C/min to 300 °C, hold 300 °C for 2 min

Method 2: Hold 75 °C for 7 min, ramp 25 °C/min to 300 °C, hold 300 °C for 2 min

Retention times for HMF and DFF in Method 1 are 6.1 and 4.4 min, respectively, and 10.5 and 8.2 min, respectively, in Method 2. Concentrations were determined by comparison of peak area to a standard curve. Supplementary Fig. 7 shows the calibration curves for HMF and DFF.

From the aqueous fraction, a 10 μL aliquot was combined with 75 μL acetonitrile and 415 μL $H_2O$ and then centrifuged and transferred to a filter vial for analysis by HPLC as described above.

**10 mL scale biotransformations**. The general scaled up biotransformation (10 mL) contained 0.05 g/L purified GOase variant, 1.7 U/mL (0.0064 g/L) HRP, 880 U/mL (0.08 g/L) catalase, and 250 mM HMF in 100 mM NaPi pH 7.4. Deviations from this are described in the text and include changes in enzyme loading, enzyme formulation, miscible or immiscible cosolvent, substrate loading, and substrate purity. After addition of HMF to start the reaction, tubes were shaken at 250 rpm at the temperatures described, however, 20 °C was typically used. Reactions were opened every hour (up to 8 h) to refresh oxygen in the headspace if no time point sample was needed at the given duration.

Aliquots (10 μL) from reactions with water miscible solvents were removed at hourly time points (1–8 h) and after 24 h and quenched by addition of 150 μL acetonitrile and 10 μL 0.5 M $H_2SO_4$ before addition of 830 μL $H_2O$. Samples were centrifuged and transferred to a filter vial for analysis. Samples were analysed by HPLC as described above.

Reactions with water immiscible solvents were quenched by addition of 1.5 mL 0.5 M $H_2SO_4$. The sample was vortexed, centrifuged, and the organic layer was removed to dry over $Na_2SO_4$. The aqueous phase was extracted twice with 20 mL ethyl acetate, combining all organic layers over $Na_2SO_4$. From the aqueous fraction, a 10 μL aliquot was combined with 37.5 μL acetonitrile and 202.5 μL $H_2O$ and then centrifuged and transferred to a filter vial for analysis. A 250 μL aliquot of the organic layer was dried by GeneVac, and the resulting solid was dissolved in 750 μL acetonitrile. The sample was prepared for analysis by combining 37.5 μL of this with 37.5 μL acetonitrile, 5 μL 0.5 M $H_2SO_4$ and 420 μL $H_2O$, centrifuged and transferred to a filter vial for analysis. Samples were analysed by HPLC as described above.

**Production of CFE for 0.2 L reactions**. GOase variants $M_{3-5}$, $M_{6-A}$ and codon optimized $M_{7-2A}$ ($M_{7-2A}^*$) were transformed and expressed as described above. Harvested cell pellets were resuspended in 50 mM NaPi pH 7.0 (5 mL per 1 g cell paste) and sonicated at 1 min on (16 amps), 1.5 min off for 6–8 cycles. The suspension was centrifuged at 39,000 × g for 30 min at 4 °C (Beckman Coulter Avanti J-E), and the clarified lysates were flash frozen in liquid $N_2$ and lyophilized. SDS gel analysis of the CFE preparations indicates similar levels of expression for all three variants (Supplementary Fig. 8).

**Specific activities of GOase CFE**. Specific activity for the three CFE preps was performed as described above for Kinetics, with 0.05 mM $CuSO_4$ in the reaction mix and using 25 mM HMF (final concentration) as substrate. Initial rates were normalized to concentration of CFE in the reactions and are presented in Supplementary Table 18.

**0.2 L scale reactions**. All 0.2 L scale biocatalytic reactions were performed in a 250 mL bioreactor (MiniBio with my-Control from Applikon Biotechnology, Delft, Netherlands). The reactor was equipped with a metal sparger, two Rushton turbines, temperature control, a dissolved oxygen sensor and a condenser to minimize stripping of the organic solvent layer. Samples were analysed on a Thermo Fisher Scientific Dionex Ultimate 3000 HPLC with a Diode Array Detector. Analytes were separated on a Bio-Rad Aminex HPX-87H column at 60 °C using an isocratic method of 15% acetonitrile/85% 5 mM sulfuric acid as mobile phase. HMF and DFF were analysed at both 230 and 280 nm, and the retention times were 20.4 min and 23.4 min, respectively.

All the concentrations refer to the concentration in the aqueous phase. Reactions were performed using 100 mM NaPi pH 7.4 as buffer and DEC as cosolvent, 0.05 mM $CuSO_4$, and 4 U/mL HRP, with the temperature kept consistent at 20 °C for all experiments. Concentrations of GOase CFE, HMF and catalase, in addition to stirring speed and aeration rate were varied as indicated to optimize performance and compare activity of GOase variants.

Reactions were performed at a working volume of 200 mL total, split between 110 mL aqueous phase and 90 mL organic phase. Catalase was added to 109 mL NaPi buffer in the reactor followed by 90 mL of HMF in DEC. A temperature sensor was added and a dissolved oxygen (DO) sensor was calibrated then added to the reactor. The DO sensor was used to monitor the activity of GOase over the reaction. The reactor was set to the desired experimental parameters (temperature, stirring speed, air flow rate into the reactor) and allowed to equilibrate. In a separate tube, the specified amount of GOase variant CFE was resuspended in 1 mL NaPi buffer, then $CuSO_4$ and HRP were added, mixed and allowed to stand for 1–2 min to activate GOase before transferring the solution to the reactor to start

the reaction. The amount of GOase CFE used in each experiment was adjusted to be equivalent to 0.05 g/L purified enzyme based on comparison of specific activity of the corresponding CFE preparation.

Samples of the reaction mixture (>1 mL) were taken at the indicated timepoints via syringe. The sample was vortexed then 950 μL was transferred to a new tube containing 50 μL 2 M sulfuric acid and vortexed to quench the reaction. After filtering (0.45 μm filter) the sample to remove emulsions and separating the phases, 10 μL was taken from both the organic and aqueous phases and each diluted with 427.5 μL 15% acetonitrile in water and 62.5 μL 2 M sulfuric acid. A portion of this sample was transferred to an HPLC vial for analysis on HPLC (5 μL injection) as described above.

**Reporting summary**. Further information on research design is available in the Nature Research Reporting Summary linked to this article.

## Data availability
Source data are provided with this paper.

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

## Acknowledgements

The research leading to these results has received funding from the European Union's Seventh Framework Programme for research, technological development and demonstration under grant agreement no. 613849 supporting the project BIOOX. The authors would also like to thank James Marshall (University of Manchester) and Darren Cook and Simon Charnock (Prozomix Ltd.) for assistance in CFE production, as well as Derren Heyes (University of Manchester) for access to equipment to perform the low oxygen screening assay.

## Author contributions

All authors contributed to the planning of the work. W.R.B. performed enzyme engineering, library screening, enzyme production and purification, kinetic characterization, and small scale biotransformations. A.T.P. performed enzyme characterization by TiTR and solvent vapour pressure calculations. M.D.G. and M.B.M. performed 0.2 L scale reactions. M.B. provided substrate formulations for use in reactions. J.M.W. and N.J.T. supervised the work. Initial draft of the manuscript was written by W.R.B. with subsequent contributions from A.T.P., M.D.G., M.B.M., M.B., J.M.W. and N.J.T.

## Competing interests

The authors declare no competing interests.
