## [Peer Review File · Nature Communications]

Reviewers' Comments:

Reviewer #1:

Remarks to the Author:

As I have indicated in the previous review-stage in Nature Catalysis, the work carried out is dedicated on the bioprocess and optimization of the reaction conditions of the different variants of galactose oxidase, resulting in high conversion yields using high concentrations of pure or crude HMF preparations underpinning the industrial potential of the target biocatalyst. This is a milestone in the field of Biocatalysis for a sustainable future in the production of biobased polymers. The authors responded well to the Reviewers' comments and suggestions and improved considerably their manuscript that is clear and well justified. The publication of this manuscript will also underpin the need for future work in improving the stability of galactose oxidase or in finding an alternative reaction system that will be further simple and economical for industrial exploitation. In my opinion, this is an excellent paper and deserves to be published in Nature Communications journal.

Reviewer #2:

Remarks to the Author:

Authors addressed most of comments from another Springer journal submission, which I have originally reviewed, so I recommend acceptance, based, of course, on the consensus of others involved. Topic is timely, well covered both from scientific as well as more applied angle, using selective enzymatic turnover.

Reviewer #3:

Remarks to the Author:

Authors have responded to the points raised in my review of an earlier submission of this manuscript. However, my main concerns remain. The evidence from the study does not support the strong claims made.

The study used enzyme and process engineering done largely in parallel, not in an integrated manner. Scalability of the enzymatic synthesis is not clear, as typical problems with retention of synthetic efficiency in agitated multiphase reactors show. Improvement by enzyme engineering in reactivity with HMF and O₂ is not clear. The previously reported enzyme variant M3-5 is similar in many relevant parameters (k_{cat} , K_m HMF, k_{cat}/K_m) to the new variants obtained in the current study. If the aim is to describe an efficient method of enzyme engineering for improved reactivity at low oxygen tension, the authors should consider to describe it in a dedicated account. The last sentence of the Abstract is not clear. It seems to combine exaggeration with vague writing. If the current study just offers a process concept or a platform, it is not novel. The manuscript conveys the idea that, through systematic development of enzyme and process, the conversion efficiency was made to exceed critical limits for an economic synthesis of 2,5-dimethylfuran from HMF. However, targets of efficiency (process metrics in Table 1) are just taken from general reviews and thus remain weakly defined. Process-specific detailed analysis is required to define targets. It is therefore never made plain where in the study they have made important breakthroughs. A journal specializing on sustainable process development (e.g., Green Chemistry) seems to be the much better venue for publication of the manuscript (revised suitably for tempered claims made).

General

Table 1 is not suitable for process evaluation. The authors seem to have taken these numbers from earlier reviews. A process-specific analysis is necessary to set the targets. A yield of around 60% may be too low by far and the use of three enzymes in a reaction cascade may simply be

unrealistic for synthesis of a bulk chemical. Moreover, various target numbers of Table 1 are not met, without discussion in the manuscript. Interpretation of enzyme TTN requires validated context from enzyme production and use in the reaction. Use of enzymes for the industrial manufacturing of a bulk chemical may require (mol-based) TTNs by far larger than about a 1 million.

Enzyme engineering

a) The enzyme variant M3-5 was previously used for HMF conversion in a 2015 Green Chemistry paper from some of the authors. Figure 1B in that paper shows complete conversion of 50 mM HMF into 2,5-dimethylfuran. I agree with the authors on the improvement made as regards the technical quality and the initial concentration of the HMF used. But, referring to the Abstract, it is difficult to see the novel "process concept", the fundamental "groundwork" or the "efficient platform".

b) The known variant M3-5 is nearly as good as the best variants reported from the current study (Tables 2 and 3). The advance made in enzyme engineering is not clear. The plate based assay for screening is from a 2008 paper. Doing the plate incubation in an environment of low oxygen tension is an interesting idea. However, to show that the procedure is truly effective, a reference/control experiment would have to be done at high oxygen tension. According to the suggestion of the authors, the reference should yield enzymes with different characteristics. After all, the M3-5 variant seems to have been obtained from a screening procedure that did not involve limitation in oxygen.

c) Although the authors use several arguments in support of low K_m for oxygen, experiments appear to have never been performed at controlled dissolved oxygen as an important process variable. Depending on the aeration used, the oxygen can vary broadly, with consequent effect on the enzyme-catalyzed rate.

Process engineering

The work done under the title "Process engineering" is a confusing mix of things that do not combine to a coherent whole. Temperature is decreased from 30°C to 20°C without remark about how this changes the enzyme activity. Organic solvent is used under widely varying conditions of agitation and the phase ratios seem to not have been constant. Aeration also varies. Different enzyme variants are used and criteria for further enzyme selection are not clear. The important effect of stability on conversion cannot be ignored at this point. At the end, a "preliminary reaction testing" is performed. This was performed at catalase loadings different from the earlier experiments. Increased catalase benefits the conversion, suggesting issues of enzyme stability that were not considered. In summary, the enzymatic process appears to be governed by various, probably interconnected factors that have not yet been identified.

Biocatalytic conversion of 5-hydroxymethylfurfural by galactose oxidase: Toward scalable technology using coordinated reaction and enzyme engineering

William R. Birmingham, Asbjørn Toftgaard Pedersen, Mafalda Dias Gomes, Mathias Bøje Madsen, Michael Breuer, John M. Woodley, Nicholas J. Turner

REVIEWER COMMENTS:

Reviewer #1 (Remarks to the Author):

As I have indicated in the previous review-stage in Nature Catalysis, the work carried out is dedicated on the bioprocess and optimization of the reaction conditions of the different variants of galactose oxidase, resulting in high conversion yields using high concentrations of pure or crude HMF preparations underpinning the industrial potential of the target biocatalyst. This is a milestone in the field of Biocatalysis for a sustainable future in the production of biobased polymers. The authors responded well to the Reviewers' comments and suggestions and improved considerably their manuscript that is clear and well justified. The publication of this manuscript will also underpin the need for future work in improving the stability of galactose oxidase or in finding an alternative reaction system that will be further simple and economical for industrial exploitation. In my opinion, this is an excellent paper and deserves to be published in Nature Communications journal.

Thank you for your positive feedback on our submission.

Reviewer #2 (Remarks to the Author):

Authors addressed most of comments from another Springer journal submission, which I have originally reviewed, so I recommend acceptance, based, of course, on the consensus of others involved. Topic is timely, well covered both from scientific as well as more applied angle, using selective enzymatic turnover.

Thanks for your review of our manuscript.

Reviewer #3 (Remarks to the Author):

Authors have responded to the points raised in my review of an earlier submission of this manuscript. However, my main concerns remain. The evidence from the study does not support the strong claims made.

The study used enzyme and process engineering done largely in parallel, not in an integrated manner. Scalability of the enzymatic synthesis is not clear, as typical problems with retention of synthetic efficiency in agitated multiphase reactors show. Improvement by enzyme engineering in reactivity with HMF and O₂ is not clear. The previously reported enzyme variant M3-5 is similar in many relevant parameters (*k*_{cat}, *K*_mHMF, *k*_{cat}/*K*_m) to the new variants obtained in the current study. If the aim is to describe an efficient method of enzyme engineering for improved reactivity at low oxygen tension, the authors should

consider to describe it in a dedicated account. The last sentence of the Abstract is not clear. It seems to combine exaggeration with vague writing. If the current study just offers a process concept or a platform, it is not novel. The manuscript conveys the idea that, through systematic development of enzyme and process, the conversion efficiency was made to exceed critical limits for an economic synthesis of 2,5-dimethylfuran from HMF. However, targets of efficiency (process metrics in Table 1) are just taken from general reviews and thus remain weakly defined. Process-specific detailed analysis is required to define targets. It is therefore never made plain where in the study they have made important breakthroughs. A journal specializing on sustainable process development (e.g., Green Chemistry) seems to be the much better venue for publication of the manuscript (revised suitably for tempered claims made).

Thank you for your review. The comments raised in this summary are addressed below, as they are provided again with more detail by the referee.

General

Table 1 is not suitable for process evaluation. The authors seem to have taken these numbers from earlier reviews. A process-specific analysis is necessary to set the targets. A yield of around 60% may be too low by far and the use of three enzymes in a reaction cascade may simply be unrealistic for synthesis of a bulk chemical. Moreover, various target numbers of Table 1 are not met, without discussion in the manuscript. Interpretation of enzyme TTN requires validated context from enzyme production and use in the reaction. Use of enzymes for the industrial manufacturing of a bulk chemical may require (mol-based) TTNs by far larger than about a 1 million.

We agree that product- and process-specific analyses would be the best way to set production metrics and targets for this study. However, specific process metrics from internal evaluation by our partner, BASF, could not be shared due to necessary confidentiality. As such, we used generic process targets for bulk chemical production as a framework for setting our targets within this academic study toward developing reaction conditions that could be fed into process development at increased scale. We considered these to be ambitious targets for an academic study, but certainly not 'final' metrics to establish industrial production of DFF. We have adjusted the presentation of this at the beginning of the manuscript to describe the selected metrics as 'proof of principle' to be used to gauge our progression and identify areas still in need of further improvement.

It is true that not all target metrics given in Table 1 were met. This was indeed mentioned in the second paragraph of the conclusion, including specific points that still need to be addressed along with potential future work of how to address these parameters. A section on this from the manuscript (with minor new revisions) is shown below, with a few additional points also being raised in the conclusions.

'Despite these advances, there are still several aspects that need further work to reach and surpass the performance metrics outlined in Table 1, in addition to a variety of considerations for translation to industrial scale. For instance, further improvements in enzyme stability are necessary to enable high conversion at increased substrate loadings to elevate production titres to meet expected production scale demands. These could potentially be found through additional enzyme engineering to improve aldehyde tolerance and biocatalyst lifetime, for example, but ultimately process development at a scale that

would allow in situ product removal (ISPR)⁷⁰ is likely required to bring productivity and specific yield into a more viable range. ’

We agree that a TTN of 1×10^6 in itself does not guarantee an industrially relevant biocatalyst, but is instead useful as a measure of productivity. The required TTN is certainly influenced by the value of the compound and the scale at which it is being produced, as well as other factors such as the efficiency of production of the biocatalyst. For production of DFF specifically, the TTN (and/or productivity metrics) may need to be higher than the 1×10^6 that we found here. However, there are not many biocatalysts that have reached this TTN threshold, which is why we wanted to highlight this quality of our evolved GOase variant.

Enzyme engineering

a) The enzyme variant M3-5 was previously used for HMF conversion in a 2015 Green Chemistry paper from some of the authors. Figure 1B in that paper shows complete conversion of 50 mM HMF into 2,5-dimethylfuran. I agree with the authors on the improvement made as regards the technical quality and the initial concentration of the HMF used. But, referring to the Abstract, it is difficult to see the novel “process concept”, the fundamental “groundwork” or the “efficient platform”.

We have softened this presentation in the Abstract, as well as throughout the manuscript, now referring to our work as ‘reaction’ development/engineering rather than ‘process’, and have used ‘coordinated’ or ‘cooperative’ rather than ‘integrated’ to describe the relationship connecting the enzyme engineering aspect and the reaction engineering aspect. We realize that ‘integrated’ can also have a technical connotation in addition to our intended use in describing the link between the two aspects of our work, and so have revised this to avoid misrepresentation of our work.

b) The known variant M3-5 is nearly as good as the best variants reported from the current study (Tables 2 and 3). The advance made in enzyme engineering is not clear. The plate based assay for screening is from a 2008 paper. Doing the plate incubation in an environment of low oxygen tension is an interesting idea. However, to show that the procedure is truly effective, a reference/control experiment would have to be done at high oxygen tension. According to the suggestion of the authors, the reference should yield enzymes with different characteristics. After all, the M3-5 variant seems to have been obtained from a screening procedure that did not involve limitation in oxygen.

The improvement in k_{cat} among the GOase variants presented here is not very large, being 2-fold at most, but this is also in comparison to our starting template of what could be considered an ‘advanced variant’ in M_{3-5} , as this was previously engineered for activity on benzylic alcohols, rather than in comparison to the wild-type enzyme with an entirely different substrate preference. Additionally, limiting the comparison to strictly these parameters overlooks a large part of the paper as there is great improvement in productivity in biotransformations under increasingly intensified conditions, part of which is due to the enzyme engineering aspect since performance in increasingly strenuous reaction conditions was included in the workflow for selecting top variants from each round. While this is hinted throughout, this total improvement is specifically highlighted toward the end of the ‘Process Engineering’ section (renamed ‘Reaction Engineering’ in the newly revised draft), through a

comparison of the productivity and specific yield of the beginning and final GOase variants. This part of the main text is copied below.

'The highest productivity though was achieved with GOase M_{7-2A} at 100 g/L semi-crude HMF in DEC after 6 h (Table 6 Entry 13), reaching 12.5 g/L.h (aqueous volume only, or 6.9 g/L.h when including solvent volume) and a specific yield 1,500 g_{DFF}/g_{enzyme}. This represents a nine-fold increase in productivity with a seven-fold increase in specific yield when compared to the M₃₋₅ progenitor enzyme at 1.4 g/L.h and 218 g_{DFF}/g_{enzyme} (Supplementary Table 2 Entry 7).'

The solid phase screening assay can certainly be performed at ambient conditions with the expectation of finding hits and is how it was utilized for this manuscript when not in the glovebox. As with any enzyme engineering screen, hits from this assay would usually be expected to have properties that correspond to the selective pressure applied during the screen (such as activity on a new substrate or activity in certain reaction conditions) so it is best practice to include this selective pressure if at all possible to improve the chances of finding new variants with the desired properties. This is why we performed the assay at reduced oxygen level in the glovebox – activity at lower oxygen levels was the parameter we were specifically wanting to target for improvement. An assay performed at ambient/high oxygen levels conditions might still fortuitously produce an enzyme variant that shows improved activity at reduced oxygen levels, however all identified hits from the assay would still need to be re-assessed in a second assay (TiTR characterization of purified enzyme or another screen in a glovebox, for example) to identify *if* any variants possess this feature, which is not a guarantee since it was not part of the screening process. Screening directly in the glovebox as we did here ensures that the selective pressure to identify variants with improved low oxygen activity is present from the start, and our secondary characterization of the one hit from this screen was to determine, as presumed at the time, the *extent* of the improvement, rather than being unsure if it was present at all.

c) Although the authors use several arguments in support of low K_m for oxygen, experiments appear to have never been performed at controlled dissolved oxygen as an important process variable. Depending on the aeration used, the oxygen can vary broadly, with consequent effect on the enzyme-catalyzed rate.

Aeration and supply of oxygen has a critical effect on the rate of catalysis as reaction rates are directly proportional to soluble oxygen concentration, while oxygen transfer rates are inversely proportional to the oxygen concentration in solution. The result is the need for a biocatalyst that operates well at sub-saturation oxygen concentration since the inherent limitation of an oxygen-dependent enzyme (i.e. activity dependence on concentration of a poorly soluble substrate) prevents it from operating at maximum performance.

A characterization of performance at varied oxygen levels was shown via the TiTR analysis of the selected variants, allowing a comparison of the true K_{MO} of the four variants tested. Separately, the biotransformations in analytical and 0.2 L scale were performed to evaluate practical productivity of the GOase variants. We thought these would be sufficient demonstrations of the improvements, rather than performing multiple 0.2 L scale reactions for each different variant and attempting to control the dissolved oxygen concentration at different sub-equilibrium (that is, below the equilibrium reached between the consumption of oxygen by GOase against the oxygen transfer rate) in each reaction. While certainly

relevant, we felt that this series of experiments was outside the scope of the current investigation.

Process engineering

The work done under the title “Process engineering” is a confusing mix of things that do not combine to a coherent whole. Temperature is decreased from 30°C to 20°C without remark about how this changes the enzyme activity. Organic solvent is used under widely varying conditions of agitation and the phase ratios seem to not have been constant. Aeration also varies. Different enzyme variants are used and criteria for further enzyme selection are not clear. The important effect of stability on conversion cannot be ignored at this point. At the end, a “preliminary reaction testing” is performed. This was performed at catalase loadings different from the earlier experiments. Increased catalase benefits the conversion, suggesting issues of enzyme stability that were not considered. In summary, the enzymatic process appears to be governed by various, probably interconnected factors that have not yet been identified.

The ‘Process Engineering’ section (now renamed ‘Reaction Engineering’ in the newly revised draft), presents a progression of our course of optimizing and intensifying the reaction conditions for GOase through interrogating the effects of individual reaction components and conditions. New materials, whether they were engineered enzyme variants, substrate formulations, or enzyme formulations, were incorporated at the point they became available to revise the system to the most relevant conditions and were used until they were replaced with another updated material. When biotransformations were performed as a secondary screen of hits from a given round of enzyme engineering, the variant with the greatest conversion in the shortest time (at the then current conditions) was selected as the top hit of the round. Much of this was quickly mentioned in the first paragraph of the section to describe our workflow, however more text has been added to clarify this.

As shown in Supplementary Table 9, performance of the M₄ variant was tested at several temperatures to identify the best reaction temperature for biotransformations. From this, the reaction at 20°C was found to provide the greatest conversion in the shortest time as an indication of a positive effect on the biocatalyst, so the standard reaction temperature for biotransformations was dropped from 25°C to 20°C. 30°C was tested as one of these conditions but was otherwise not used for biotransformations.

The cosolvent used varied as 5 or 10% for water miscible solvents and 80% (of aqueous volume) total of water immiscible solvents, in the first instance as two additions of equal volumes, as described in Supplementary Table 7 and previous tables in the SI during the solvent screening and optimization experiments. Once EtOAc was selected from this initial screen, the final ratio was kept consistent throughout the rest of the work (analytical, 10 mL and 0.2 L scales) as one addition at the start of the reaction to simplify the initial two step addition of solvent, with only the identity of the solvent changing. The aeration rates (shaking/stirring speed and volume gas/volume reactor/min (vvm) rates) were adjusted according to results collected. Analytical scale biotransformations were all performed at 250 rpm. The 0.2 L scale biotransformations were initially performed at 1000 rpm and 1 vvm air but this led to a high degree of foaming, as described in the corresponding Supplementary Note, and both rates were consequently reduced for the experiments provided in Supplementary Table 17 (500 rpm stirring and 0.09 vvm air).

The requirement of catalase to remove H_2O_2 has been previously reported, and a demonstration of its deleterious effects presented in our article from 2015 titled 'Process requirements of galactose oxidase catalyzed oxidation of alcohols' (*Org. Process Res. Dev.* 19, 1580-1589). The catalase loading in the 0.2 L reaction was initially high, which contributed to the foaming issue discussed in the Supplementary Note on these reactions. It was drastically lowered as one of the several measures implemented to reduce foaming, and then the concentration was adjusted upward to observe the effects on foaming as well as conversion. Foaming was not a concern in the shaken analytical scale reactions, so this aspect could only be investigated at the larger scale reactions and is still an aspect that needs attention during a translation to larger volume reactions.

Reviewers' Comments:

Reviewer #1:

Remarks to the Author:

I agree with the changes made by the authors on the concerns indicated by the reviewers, adjusting the manuscript accordingly. The work done is important indicating the development of a biocatalyst showing strong potential for the industrial production of biobased DFF. I believe that the manuscript is ready to be published in Nature Communication journal that will be of great interest for the scientific community!

Toward scalable biocatalytic conversion of 5-hydroxymethylfurfural by galactose oxidase using coordinated reaction and enzyme engineering

William R. Birmingham, Asbjørn Toftgaard Pedersen, Mafalda Dias Gomes, Mathias Bøje Madsen, Michael Breuer, John M. Woodley, Nicholas J. Turner

REVIEWERS' COMMENTS

Reviewer #1 (Remarks to the Author):

I agree with the changes made by the authors on the concerns indicated by the reviewers, adjusting the manuscript accordingly. The work done is important indicating the development of a biocatalyst showing strong potential for the industrial production of biobased DFF. I believe that the manuscript is ready to be published in Nature Communication journal that will be of great interest for the scientific community!

Thanks very much! We appreciate your comments toward strengthening our manuscript.